# High-Tech Home-Based Rehabilitation after Stroke: A Systematic Review and Meta-Analysis

**DOI:** 10.3390/jcm12072668

**Published:** 2023-04-03

**Authors:** Soo-Kyung Bok, Youngshin Song, Ancho Lim, Sohyun Jin, Nagyeong Kim, Geumbo Ko

**Affiliations:** 1Department of Rehabilitation, College of Medicine, Chungnam National University, Daejeon 35015, Republic of Korea; 2Department of Nursing, College of Nursing, Chungnam National University, Daejeon 35015, Republic of Korea

**Keywords:** home, physical function, stroke rehabilitation, systematic review, virtual

## Abstract

(1) Background: To improve existing rehabilitation technologies, we conducted a systematic review and meta-analysis to identify the effect size of home-based rehabilitation using robotic, virtual reality, and game devices on physical function for stroke survivors. (2) Methods: Embase, PubMed, Cochrane Library, ProQuest, and CINAHL were used to search the randomized controlled trials that applied technologies via home-based rehabilitation, such as virtual reality, robot-assisted devices, and games. The effect size (Hedges’s g) of technology type and affected limb on physical function were calculated. (3) Results: Ten studies were included. The effect size of home-based rehabilitation in virtual reality had the greatest value (Hedges’s g, 0.850; 95% CI, 0.314–1.385), followed by robot-assisted devices (Hedges’s g, 0.120; 95% CI, 0.003–0.017) and games (Hedges’s g, −0.162; 95% CI, −0.036 to −0.534). The effect size was larger in the upper limbs (Hedges’s g, 0.287; 95% CI, 0.128–0.447) than in the lower limbs (Hedges’s g, −0.113; 95% CI, −0.547 to 0.321). (4) Conclusions: Virtual reality home rehabilitation was highly effective for physical function compared to other rehabilitation technologies. Interventions that consisted of a pre-structured and tailored program applied to the upper limbs were effective for physical function and psychological outcomes.

## 1. Introduction

Stroke, or cerebrovascular accident (CVA), is one of the most common causes of disability globally. The focus of stroke treatment is the restoration of blood flow to the brain and the control of secondary neurological damage [1]. Stroke has a negative impact not only on the individual’s physical health, but also on the psychological, social, and emotional health, depending on the severity, and it increases the burden on caregivers, including family members [1,2,3,4]. Rehabilitation is very important to this process. Intensive rehabilitation programs focus on functional recovery due to long-lasting physical impairments associated with stroke [1,2]. As an alternative to inpatient rehabilitation, home-based rehabilitation allows individuals to tailor their program to meet their preferences and schedules [3]. Although face-to-face home-based rehabilitation provides individualized services via physical consults from the rehabilitation team, it remains impersonal and a cost burden to both clinicians and patients [4]. With recent advances in technology and the increasing preference for virtual consultations due to the coronavirus disease 2019 (COVID-19), home-based virtual technologies are adapting to promote functional abilities [3,5]. Home-based rehabilitation support has been provided mainly through telecommunication devices such as videophones and telephones; however, virtual reality (VR) or robot-assisted devices (RD) have recently been introduced to facilitate home-based rehabilitation for stroke survivors [5]. These home-based technologies aim to improve patients’ physical function and internal motivation by encouraging their progress and goal attainment. Although several systematic reviews have evaluated the effect of new home-technologies on patient clinical outcomes, positive results were not always guaranteed [6,7,8]. For example, Maier et al. reported that the effect of VR on motor function recovery after stroke is unclear [7], and Hatem et al. reported that task-oriented RD therapy and VR were not recommended to enhance physical recovery in the subacute stage (<6 months) after stroke [6]. However, most of these conclusions were based on insufficient scientific data. Chen et al. also reviewed 31 articles that utilized a technology-based home rehabilitation service and defined 6 types of technologies (games, robotics, virtual reality, etc.), but they did not propose scientific parameters, such as effect size, for each technology [5].

A study also reviewed the effect of rehabilitation intervention according to the target limb. In a review by Hatem et al. that evaluated the effects of various rehabilitation interventions on upper limb function after stroke, RDs and VR were integrated as adjuvant therapies into stroke because of a lack of evidence even in mirror therapy [6]. Another limitation of this review was that they focused solely on upper limb function without focusing on the program delivery method or the effect on any other body part.

As such, the effectiveness of home-based rehabilitation may vary depending on the types of technology for rehabilitation, intervention contents, and target limb. Thus, this study aimed to identify the effect size of different technologies by systematically reviewing and analyzing studies that applied technology in home-based stroke rehabilitation programs. Our research questions are as follows:(a)What are the effect sizes of VR, RDs, and games in home-based rehabilitation?(b)Does the effect size differ depending on the type of technique applied?(c)Does the effect size on the physical function of the stroke survivor differ depending on the target limb?

## 2. Materials and Methods

This systematic review and meta-analysis followed the Preferred Reported Items for Systematic review and Meta-Analysis (PRISMA) guidelines (PRISMA checklist; Appendix A), but the review was not registered.

Literature search strategy: Two research groups independently searched and analyzed the literature using the same strategy (years of publication, English language, search terms, and database). Literature published from the date of inception of the database to 28 February 2021, was systematically searched using electronic databases, including PubMed, Cochrane Library, CINAHL, Embase, and PsyINFO. Ultimately, Google Scholar was searched to identify missing research papers. The selected literature was organized using reference management software (EndNote 20). The following search terms were used: MeSH terms, including “stroke,” “rehabilitation,” “telemedicine,” “telerehabilitation,” “reality therapy,” and “augmented reality” were used as search keywords. That is: (((stroke [MeSH Terms] OR stroke [All Fields] OR cerebrovascular accident [All Fields] OR CVA [All Fields]) AND (rehabilitation [MeSH Terms] OR rehabilitation [All Fields])) OR stroke rehabilitants [All Fields]) AND ((telemedicine [MeSH Terms] OR telemedicine [All Fields]) OR (telerehabilitation [MeSH Terms] OR telerehabilitation [All Fields]) OR (technology [MeSH Terms] OR technology [All Fields]) OR (reality therapy [All Fields] OR reality therapy [MeSH Terms]) OR (augmented reality [All Fields] OR augmented reality [MeSH Terms]) OR digital* [All Fields] OR smart* [All Fields] OR untact* [All Fields]) AND LANGUAGE (English) AND PUBYEAR > 2009.

Eligibility criteria: Randomized controlled trials (RCTs) that evaluated stroke survivors living at home who received home-based rehabilitation technologies were included in the systematic review. Pilot and non-RCT studies were excluded. P (patient), I (intervention), C (Comparison), and O (Outcomes) were defined as follows: P (stroke survivors who are dwelling in their home), I (home-based rehabilitation that applied technologies), C (usual care or conventional care), and O (physical function).

Review and coding process: All researchers were divided into two teams for the review process. First, the abstracts were imported and checked for duplication. Second, the full text of the publication was reviewed to ensure the eligibility criteria were met. Third, once the final literature was confirmed, the characteristics of each study were coded by both teams. The coding scheme included the authors, year of publication, study design, number of participants, intervention details (type and duration of intervention, inclusion criteria, primary outcomes, and measurement tools), and research findings. For subgroup analysis, the target limbs were classified into upper and lower based on a prior study [5]. The types of primary outcomes were coded according to the outcomes measured. If discrepancies were found among codes, the principal investigator had the final judgment after discussing it with the other researchers.

Types of intervention: According to the review by Chen et al., we classified the interventions into four types for subgroup analysis: VR (interactive games or virtual exercises), games (piano, fishing, sport games, etc.), and RD (facilitates movement of the arm, wrist, and hands to improve the range of motion). Studies were excluded that applied telecommunication devices, such as mobile and video conferencing, to assist patients in remotely accessing medical services from health providers [5].

Target limb: The target limb for intervention was divided into upper and lower limbs.

Types of primary outcomes: To classify the interventions according to primary outcomes, the tools used for measurement were analyzed. Studies that used the following outcome measures were coded as physical function outcomes: Fugl–Meyer Assessment (FMA), Wolf Motor Function Test (WMFT), Nine-Hole Peg Test (9HPT), Box and Block Test, Action Research Arm Test (ARAT), Manual Function Test (MFT), Purdue Pegboard Test, Stroke Impact Scale (SIS), Tinetti Performance-Oriented Mobility Assessment (POMA), 10-meter Walk Test (10MWT), and Time-Up and Go test (TUG). In addition, the following measures were coded as activity of daily living (ADL) outcomes: the Modified Barthel Index (MBI) and Nottingham Extended ADL Index. Strength was measured using standardized Manual Muscle Testing (MMT), and studies assessing the Berg Balance Scale and Brunel Balance Assess (BBA) were coded as balance outcomes. Studies that evaluated psychological factors with the Center for Epidemiologic Studies Depression Scale (CESD) were regarded as psychological outcomes.

Quality appraisal: The Cochrane risk of bias tool (ROB) was used to evaluate each study. The ROB consists of seven domains: random sequence generation (selection bias), allocation concealment (selection bias), blinding of participants and personnel (performance bias), blinding of outcome assessment (detection bias), incomplete outcome data (attrition bias), selective reporting (reporting bias), and other biases. Each domain is answered as either “yes” (low risk of bias), “no” (high risk of bias), or “uncertain” (unclear risk of bias). The research teams evaluated and discussed the quality of the studies according to these domain criteria to determine the final risk of bias.

Statistical analysis: The ROB was analyzed and visualized using the Review Manager (RevMan) program (Cochrane Collaboration; London, UK). The meta-analysis was performed using the Comprehensive Meta-Analysis 3.0 (CMA, Biostat; Englewood, NJ, USA) program. Hedges’s g was measured to evaluate effect size, which is generally preferred to Cohen’s d. Hedges’s g recommends classifying 0.2 for a small effect, 0.5 for a medium effect, and 0.8 for a large effect [9].

Hedges’s g formula is
Hedges’s g = (M1 − M2)/SD_pooled_(M1 − M2: mean difference; SD_pooled_: pooled and weighted standard deviation).

Statistical heterogeneity was assessed using the I^2^ index to quantify the degree of heterogeneity [9,10]. Percentages around 25% (I^2^ = 25), 50% (I^2^ = 50), and 75% (I^2^ = 75) would indicate low, medium, and high heterogeneity, respectively. We considered that the statistical heterogeneity (I^2^) and clinical implication of studies for effect sizes could indicate whether a random-effects or fixed-effects model should be selected [9,10,11]. The random-effects model was applied in this meta-analysis.

We also examined Q-values between study groups to determine whether the effect size differed significantly among intervention types and target limbs. If the Q-value was statistically significant (*p* < 0.001), the effect size was different between study groups. Publication bias, or reporting bias, is a bias that states research with meaningful results is more likely to be published. Fail-safe N was calculated to estimate the number of missing studies required for statistical significance [9].

## 3. Results

### 3.1. Characteristics of the Included Studies

The initial database search yielded 9540 records. We screened the titles and abstracts of 9006 non-duplicate records and excluded 8655 articles not relevant to the topic. Therefore, 351 full-text articles were reviewed for eligibility; 50 studies meeting the search criteria were included in the qualitative analysis, of which 10 were used in the meta-analysis. The detailed results of our research strategy are presented in Figure 1 as a PRISMA flowchart.

The characteristics of the included studies (*n* = 10) are presented in Table 1. Among the intervention types, 3 studies used VR, 3 used games, and 4 used RD rehabilitation. The total number of participants was 761 (experimental, 383; control, 378). The mean ages of the experimental and control groups were 60.4 and 59.3 years, respectively. The duration of the interventions ranged between 3 and 10 weeks (Table 1).

### 3.2. The Risk of Each Study

As mentioned above, the quality of each study was assessed according to the ROB tool’s seven domains for RCTs recommended by the Cochrane Collaboration. In the first domain (random sequence generation), all 10 included studies were rated as having a low risk of bias. Concerning allocation concealment, eight studies were rated as a low risk of bias, but two were unclear. Six studies were evaluated as having a low risk of bias for the blinding of participants and personnel domain, and eight studies had a low risk of bias for the blinding of the outcome assessment domain. In the incomplete outcome data domain, eight studies were found to have a low risk of bias. Lastly, in the selective reporting domain, eight studies were assessed to have a low risk of bias (Figure 2).

### 3.3. Publication Bias

To obtain information regarding publication bias, we calculated a Fail-Safe N (FSN) to determine the number of missing studies required to nullify this effect. The result showed the FSN to be 164 for the systematic review and 37 for the meta-analysis, indicating we would need to locate and include 164 “null” studies for the combined 2-tailed p-value to exceed 0.05. Therefore, this meta-analysis had no publication bias.

### 3.4. Effect Size by Types of Intervention

The VR intervention showed the largest effect size (Hedges’s g, 0.850; 95% CI, 0.314–1.385, I^2^ = 77.5%, Q-value = 35.7 (*p* < 0.001)), followed by robot-assisted interventions (Hedges’s g, 0.129; 95% CI, 0.025–0.232, I^2^ = 0%, Q-value = 12.3 (*p* = 0.015)). In contrast, the effect size of the game intervention was insignificant (Hedges’s g, −0.162; 95% CI, −0.534 to 0.210, I^2^ = 77.4%, Q-value = 22.1 (*p* < 0.001)). Heterogeneities were found in VR and game intervention studies. The effect sizes among the types of intervention were significantly different (between study; Q-value = 9.335 (*p* < 0.009)).

Among the VR studies, the effect size in the study by Standen et al. [16] was the highest (Hedges’s g 2.152) and it significantly improved the physical function, such as motor function. Among studies on robot-assisted devices, the Zondervan et al. [17] study found that applying the MusicGlove changed motor activity significantly. In game studies, Hsieh [14] found that video games significantly changed the 10MWT (Hedges’s g 0.598) and body Anterior–Posterior (AP) sway (Hedges’s g −0.913)/sway area (Hedges’s g −0.761) (Figure 3).

### 3.5. Effect Size by Applied Limb

Target limbs were used as subgroups for home-based rehabilitation after stroke. The effect size in interventions targeting the upper limb was larger (Hedges’s g, 0.291; 95% CI, 0.133–0.449, I^2^ = 62.1%, Q-value = 76.6 (*p* < 0.001)) than the lower limb (Hedges’s g, −0.113; 95% CI, −0.547 to 0.32, I^2^ = 73.3%, Q-value = 22.5 (*p* = 0.001)). Heterogeneities were found in upper and lower limb studies. The effect size was significantly different by limb (between study Q-value = 2.939, *p* = 0.086) (Figure 4).

## 4. Discussion

The purpose of this study was to identify the amount of home-based technology rehabilitation research conducted to date, the types of intervention, and their effect sizes. From this systematic review process, 10 studies were found with 761 stroke survivor participants. Most participants were aged in their 50s and 60s and between 3 months and 12 months post-stroke, although most participants were 3–6 months post-stroke. In the studies, the participant’s baseline function was defined after the acute phase of rehabilitation, whereby some degree of physical and cognitive function was recovered.

The meta-analysis found the overall effect size of home-based rehabilitation using technologies such as RDs, VR, and games to be moderately low in terms of efficacy for moderately improving physical function in stroke survivors. However, along with the development of technology in telemedicine, the number of telemedicine home-based rehabilitation studies has increased. For example, a review study by Rubin et al., who reviewed 24 studies that provided telemedicine for post-stroke care published prior to 2012, found that only 8 of the total studies took place in the home setting [8].

Since 2015, studies on VR have accumulated, with 3 studies relative to home-based rehabilitation. Of these studies, two used a virtual exercise program [16] and virtual gloves [20] for upper arm function, and a single study used virtual-reality-based exercise [13] to improve the balance for gait. VR intervention was facilitated over 8 weeks and tailored to the patient prior to discharge or adjusted by the therapist. VR, which contains game elements to arouse interest, was also applied to the program [20]. All VR studies enhanced motor function and balance scores and had the largest effect size amongst the intervention types. However, some studies noted a high dropout rate because of technical issues, suggesting the need for a medical and technical support team.

Gaming software was designed to run on a computer; the contents consisted of either driving, piano, fishing, or general sports games in three studies [12,13,15]. However, no difference in arm function between the experimental and control group was observed, only some changes in gait. As a possible explanation, studies have reported that these game programs were not developed for treating stroke survivors. Nevertheless, in Hsieh’s study, a video game using a foot switch program (developed for treatment) effectively improved participants’ walking ability [14].

Four studies using RD interventions were found, although all were focused and applied to the upper extremities [17,18,19,21]. These studies utilized an RD that facilitated the active movement of the upper limbs using customized programs and stored data. In 2 studies, the device was provided for 2 months; in 1 study, an RD was used for only 3 weeks. Nevertheless, no significant difference between the groups was observed in the upper limb function of patients. These studies were included as the treatment group in three of the four studies only provided RDs to participants, while the control group maintained interaction with the therapist (somewhat similar to telerehabilitation). Although these results are similar to the existing review studies [5,6], careful interpretation is necessary because of the variable quality of traditional care provided in the control groups.

All studies included in this meta-analysis were pre-structured and programmed studies of VR, RDs, and games. Most were customized before intervention according to the participants’ functional state, cognitive capability, and caregivers’ ability to assist. A similar effect size score was also observed in a study by Wu et al., whereby the intervention was tailored to the participants virtually by the therapists of a collaborative care team during the intervention period; as a result, the participants’ motor function increased [22]. Previous studies have reported that a team approach that includes both healthcare staff and caregivers maximizes the rehabilitation of stroke survivors [23,24,25]. These efforts eventually reduce medical costs [26]. Prior studies included self-administrated programs such as mirror therapy for home-based rehabilitation [27], but these were not included in this study. Mirror therapy, which allows participants to see images of their limbs in a mirror, works differently than VR because it requires focused individual effort [6].

During the review process, classifying individual studies according to intervention sites showed that upper limb studies were the most frequent (eight studies), followed by lower limb (two studies). The studies focused on the upper limb assessed physical function, strength, and ADLs, whereas balance and muscle strength were included in lower limb studies. For measuring physical function, the most common tools were the FMA and Box and Block test. Strength was measured mostly using MMT; MBI was most used to assess the ADLs. Moreover, the effect size of upper limb interventions was larger than that of lower limb interventions. The number of studies specific to intervention type for the lower limbs was one for games and one for VR; each consisted of remote exercises, a mobile exercise program, and a video game program. Those two studies measured balance with the BBS; walking capacity was measured using the TUG test and the 10 MWT. However, the study found significant differences between groups after lower limb game interventions. Notably, the two studies on lower extremity function selected in this study are heterogeneous, and the studies included a large amount of upper limb research. Therefore, attention is needed in interpreting the results of this meta-analysis, considering the research situation of individual studies.

This meta-analysis identified that the effect size of home rehabilitation differs by technology type and target limbs. However, some considerations are necessary when interpreting and applying the results of this study. Because of the quality evaluation, limitations such as vague explanations of interventions were found in some studies. Therefore, some discrepancies between the results of the moderator classified in this study and the actual intervention content may exist. While the expansion of home-based rehabilitation using VR, RDs, and games is expected, developing a balanced intervention between the provider’s modalities in conjunction with advancing technology seems necessary.

The strengths and limitations of this study are as follows: (a) The study search, selection, data extraction, and quality appraisal process were performed rigorously and systematically by following the PRISMA guidelines. However, PERSiST (Prisma in Exercise, Rehabilitation, Sport medicine and SporTs science), which is recently recommended as a guideline for SR research in the field of sports [28], was not used in this study. This is a limitation of this study. (b) The intervention for home-based rehabilitation had moderate–low efficacy in physical function for stroke survivors. (c) This systematic review and meta-analysis presented that the effect size of home rehabilitation differs by technology type and target limbs. (d) The content of the intervention presented in the included paper was unclear; hence, accurate technical classification was limited in terms of types of intervention. (e) Studies were searched using a limited database; hence, the possibility of missing some studies cannot be ruled out.

## 5. Conclusions

In summary, home-based rehabilitation using technology is effective in recovering participants’ physical function and ADLs when customized programs are applied to the upper limb. The quality of 10 individual studies derived from the review process was robust, and no publication bias was detected. The effect size of VR, RDs, and game rehabilitation interventions on physical function was moderately small. Interventions using VR that were either a pre-structured or a tailored program for upper limbs improved the psychological factors, physical function, and ADLs of stroke patients. This systematic review and meta-analysis can help determine strategies for home-based rehabilitation studies, participant calculations, and primary outcome selection.

## Figures and Tables

**Figure 1 jcm-12-02668-f001:**
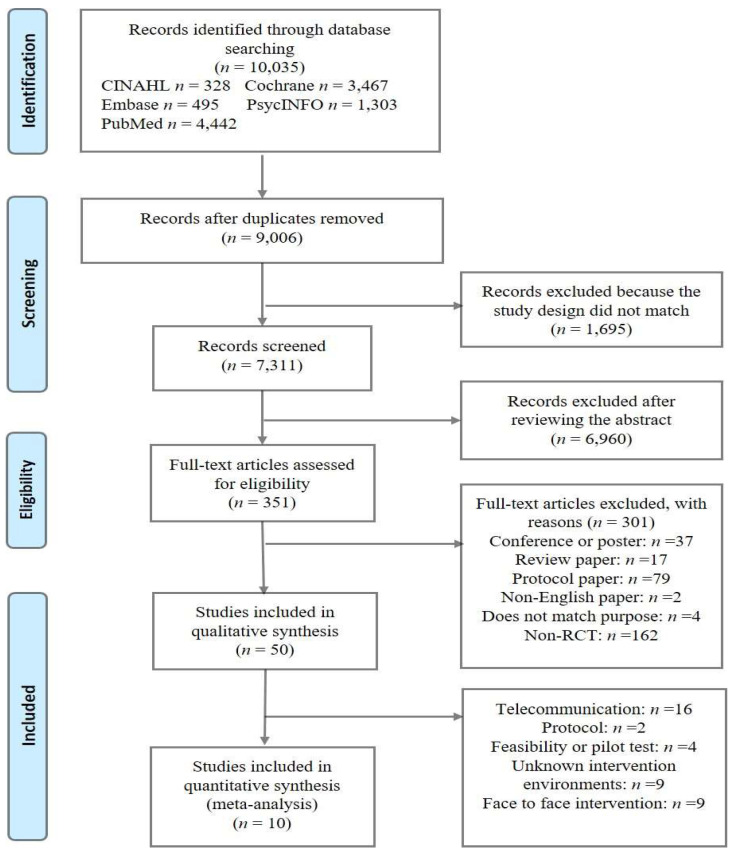
Search strategy.

**Figure 2 jcm-12-02668-f002:**
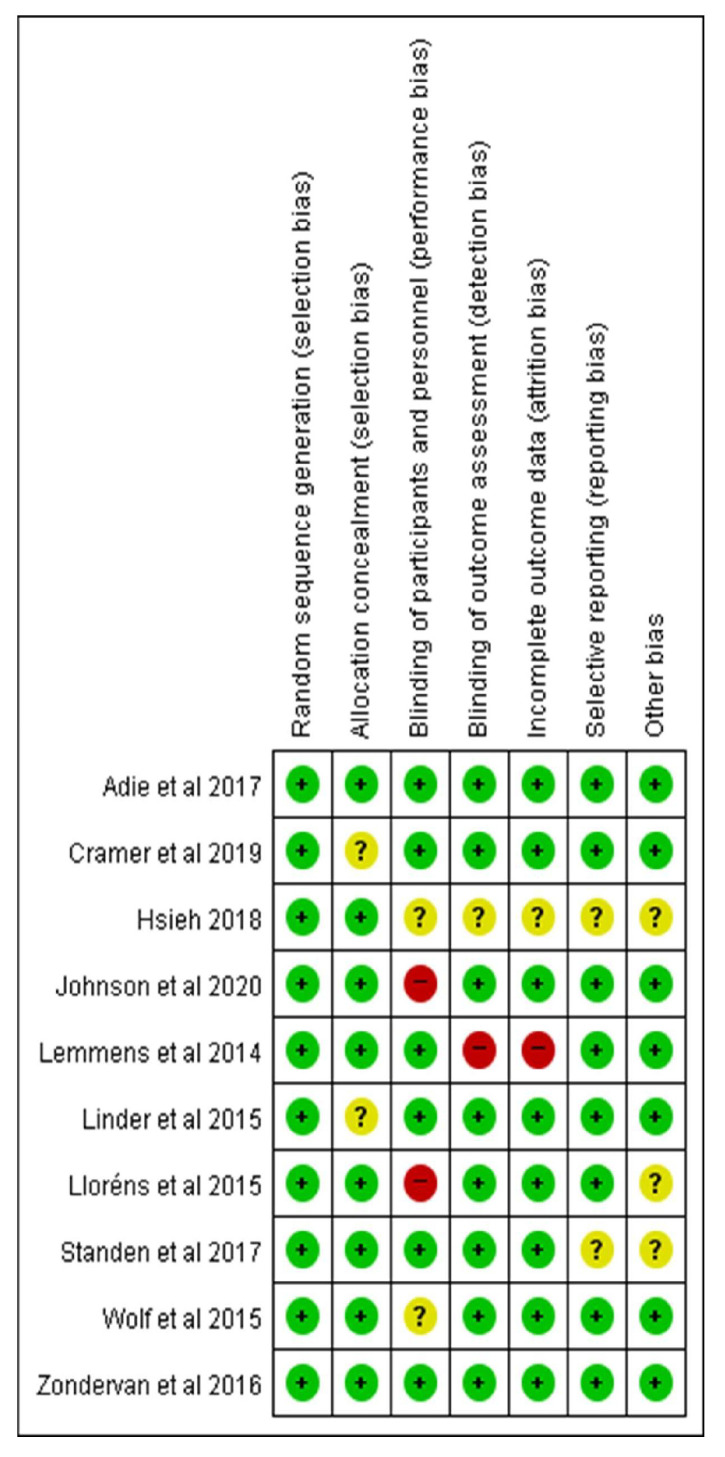
Risk of bias for 10 studies (green; low risk, yellow; unclear, red; high risk) [12,13,14,15,16,17,18,19,20,21].

**Figure 3 jcm-12-02668-f003:**
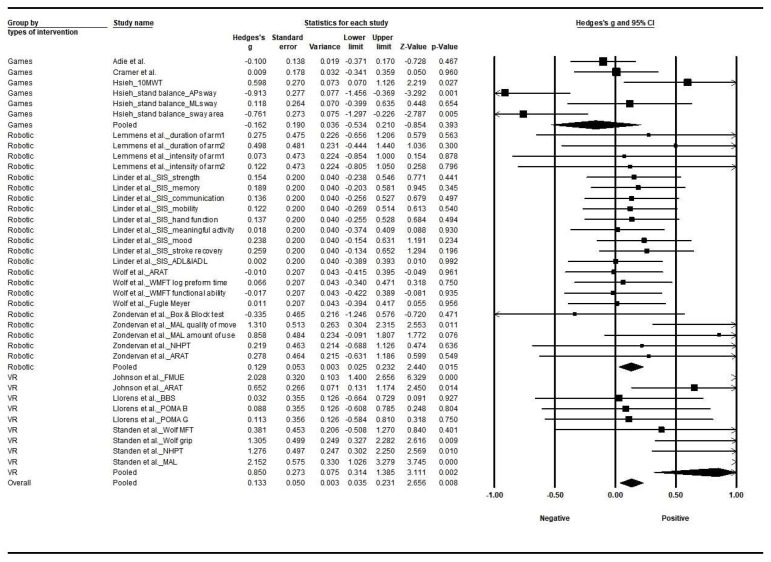
Forest plot among types of intervention. Abbreviations: 10 MWT, 10-Meter Walking Test; AP, anterior–posterior; ML, medial–lateral; SIS, Stroke Impact Scale; (I)ADL, (Instrumental) Activity in Daily Life; ARAT, Action Research Arm Test; BBS, Brunel Balance Assessment; POMA B, Performance-Oriented Mobility Assessment-Balance; POMA G, Performance-Oriented Mobility Assessment-Gait; MFT, Motor Function Test; NHPT, Nine-Hole Peg Test; MAL, Motor Activity Log. (squares and circle show a mean of effect size, arrow presents the 95% confidential interval). Summary of Figure 3: The effect size, Hedges’s g, was −0.162 (*p* = 0.393) in games [13,14,15], 0.129 (*p* = 0.015) in robot-assisted devices [17,18,19,21], and 0.850 (*p* = 0.002) in virtual reality [12,16,20]. That is, robot-assisted devices and VR intervention had a positive effect on physical function. VR, virtual reality.

**Figure 4 jcm-12-02668-f004:**
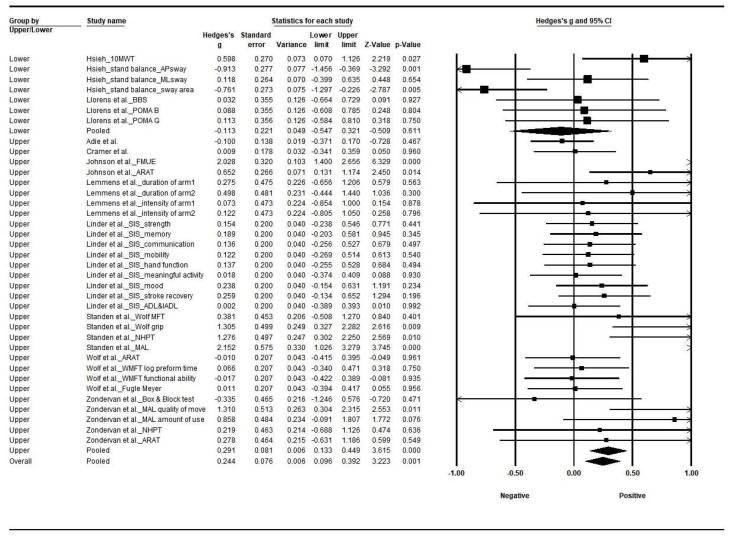
Forest plot by applied limb. Abbreviations: 10MWT, 10-Meter Walking Test; AP, anterior–posterior; ML, medial–lateral; SIS, Stroke Impact Scale; (I)ADL, (Instrumental) Activity in Daily Life; ARAT, Action Research Arm Test; BBS, Brunel Balance Assessment; POMA B, Performance-Oriented Mobility Assessment-Balance; POMA G, Performance-Oriented Mobility Assessment-Gait; MFT, Motor Function Test; NHPT, Nine-Hole Peg Test; MAL, Motor Activity Log. (squares and circle shows a mean of effect size, arrow presents the 95% confidential interval). Summary of Figure 4: The effect sizes, Hedges’s g, were −0.113 (*p* = 0.750) in the lower limbs [14,20] and 0.291 (*p* < 0.001) in the upper limbs [12,13,15,16,17,18,19,21]. The effect size in the upper limbs was significant. Games [13,14,15], VR [12,16,20], and robot-assisted interventions [17,18,19,21] applied to the upper limbs showed positive effects.

**Table 1 jcm-12-02668-t001:** Characteristics of the Ten Studies.

Author	Intervention	Interventions	Duration	Participants *n* = 761 (Mean Age)	Inclusion Criteria	Primary Outcome Measures	Main Findings
1. Johnson et al., 2020 [12]	VR	Virtual therapy: 45 min, 2 times/week	2 m	Exp = 28 (64.7 yr) Cont = 30 (59.3 yr)	≥18 yr, FMA (upper extremity) Score 25–45, MMSE > 24, at least 3 months poststroke	FMA (upper extremity), ARAT	The FMA (upper extremity) score was improved
2. Cramer et al. [13]	Games	Computer game (36 sessions) for TR group Cont: printed homework (contents the same as the TR group)		Exp = 62 (62 yr) Cont = 62 (60 yr)	18 yr old, Stroke 4 to 36 wks prior, mild to severe arm motor deficit	Arm motor (FMA, BBT)	No difference in the FMA score
3. Hsieh [14]	Games	VG intervention (foot switch): 30 min (once a week)	2.5 m	Exp = 28 (58.3 yr) Cont = 28 (59.3 yr)	Stroke onset over 3 m, ankle dorsiflexion (>10°), ability to walk	Walking performance (10MWT, CoP sway, AP sway, ML sway) and standing balance (CoP sway)	Walking performance was improved (except CoP and ML sway)
4. Adie et al. [15]	Games	Wii exercise: 45 min/daily × 6 wks	1.5 m	Exp = 117 (66.8 yr) Cont = 118 (68 yr)	Stroke within 6 m, Medical Research Council Scale ≤ 5	ARAT	No differences in all variables
5. Standen et al. [16]	VR	VR Nintendo Wii remote): 20 min × 3 times/day	2 m	Exp = 17 (59 yr) Cont = 10 (63 yr)	Residual arm dysfunction	WMFT, Nine-Hole Peg test, MAL, Nottingham Extended ADL, Frequency of use	The WMFT and MAL score were improved
6. Zondervan et al. [17]	RD	MusicGlove device and laptop: 3 wks	3 wks	Exp = 9 (60 yr) Cont = 8 (59 yr)	BBT ˃ 1, age ˂ 75	BBT	No difference in BBT score
7. Linder et al. [18]	RD	Home-based robot (Hand Mentor Pro) assisted rehab: 8 wks, 5 days a week, 3 h	2 m	Exp = 48 (55.5 yr) Cont = 51 (59.4 yr)	Stroke within 6 m, FMA 11–55	SIS, CESD	No difference in SIS and CESD scores
8. Wolf et al. [19]	RD	Home-based robot (Hand Mentor Pro) assisted rehab: 5 days/week, 3 h (total 120 h)	2 m	Exp = 51 (59.1 yr) Cont = 48 (54.7 yr)	Stroke within 6 m, FMA 11–55	ARAT	No difference in variable
9. Lloréns et al. [20]	VR	Home-based VR for balance: 45 min, 3 times/week	2 m	Exp = 15 (55.5 yr) Cont = 15 (55.6 yr)	≥40 aged ≤75, >6 m, BBA 7–12, MMSE > 23	BBS, POMA-B, POMA-G, BBA, SIS, IMI	The scores of BBS, POMA-B, POMA-G, and BBA were improved
10. Lemmens et al. [21]	RD	Robot-supported task-oriented arm-hand training: 8 wks, 4 times/week, 2 × 30 min/day	2 m	Exp = 8 (63.5 yr) Cont = 8 (55.0 yr)	18–85 aged, first-ever stroke, MRC grade 2–4, post-stroke time ≥ 12 m, MMSE ≥ 26	FMA, ARAT, MAL	No differences in both groups

(Abbreviations) ADL: Activity of Daily Living; AP sway: sway in Anterior–Posterior direction; ARAT: Action Research Arm Test; BBA: Brunel Balance Assess; BBS: Berg Balance Scale; BBT: Box and Block Test; BI: Barthel Index; CESD: Center for Epidemiologic Studies Depression Scale; Cont: Control group; CoP sway: Center of Pressure sway; Exp: Experimental group; FMA: Fugl–Meyer Assessment; IMI: Intrinsic Motivation Index; MAL: Motor Activity Log; MBI: Modified Barthel Index; MFT: Manual Function Test; ML sway: sway in Medio-Lateral direction; MMSE: Mini-Mental State Examination; MRC grade: grading of muscle power; POMA-B: Performance-Oriented Mobility Assessment-Balance Test; POMA-G: Performance-Oriented Mobility Assessment-Gait Test; RD: Robot-Assisted Device; SIS: Stroke Impact Scale; VG: Video Game; VR: Virtual Reality; WMFT: Wolf Motor Function Test; 10 MWT: 10-Meter Walking Test.

## Data Availability

Not applicable.

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
