# Peer review of "High-Tech Home-Based Rehabilitation after Stroke: A Systematic Review and Meta-Analysis"

_jcm, 2023, doi:10.3390/jcm12072668_

Round 1
Reviewer 1 Report
Please check the attached feedback for your action.

Author Response
We thank the reviewers for their encouragement and thorough critique. We have provided point-by-point responses to each reviewer’s comment below. In addition, we have revised the manuscript as advised; for clarity, these changes are highlighted in red font in the manuscript.

Reviewer 2 Report
Dear Authors,
First of all, I would like to congratulate you on the development of this research. The topic is novel and it could be more than justified to carry out a systematic review on the subject of tele-rehabilitation.
However, the manuscript presents very serious methodological limitations in the Methodology section: (a) It obviates the inclusion of very important databases in Health Sciences such as Web of Science and Scopus (among others); (b) The most appropriate checklist for conducting systematic reviews in Rehabilitation is the PERSIST adaptation of PRISMA; (c) The authors are not transparent with the equations and search filters applied in each database (this is the least serious of the methodological errors); and (d) The authors do not define all the acronyms of the PICOS.
For all these reasons, among other minor errors (grammatical and typos), I consider deeply corrected and rewritten, and subsequently resubmitted for this Journal.
Kind regards.
Author Response

(The authors gave the same response as above.)

Reviewer 3 Report
The review article by Bok et al. "High-tech device assisted home rehabilitation after stroke: Sys tematic review and meta-analysis" covers a potentially interesting and emerging topic related to the stroke rehabilitation. In this sense, this remains to be potentially interesting for the Journal of Clinical Medicine readers.
I regard the main point of this paper as highly attractive as well as the results are clearly presented. The text does not contain any major errors, therefore I have some minor comments and recommendations:
1. There is a need to provide slightly more expanded introduction shortly
mentioning/describing pathogenesis of stroke
2. The figure summarizing and clarifying the conclusions should be added.
3. Following references should be added and properly cited within the main text:
- Mela A, Poniatowski ŁA, Drop B, Furtak-Niczyporuk M, Jaroszyński J, Wrona W, Staniszewska A, Dąbrowski J, Czajka A, Jagielska B, Wojciechowska M, Niewada M. Overview and Analysis of the Cost of Drug Programs in Poland: Public Payer Expenditures and Coverage of Cancer and Non-Neoplastic Diseases Related Drug Therapies from 2015-2018 Years. Front Pharmacol. 2020 Aug 14;11:1123. doi: 10.3389/fphar.2020.01123.
- Winstein CJ, Stein J, Arena R, Bates B, Cherney LR, Cramer SC, Deruyter F, Eng JJ, Fisher B, Harvey RL, Lang CE, MacKay-Lyons M, Ottenbacher KJ, Pugh S, Reeves MJ, Richards LG, Stiers W, Zorowitz RD; American Heart Association Stroke Council, Council on Cardiovascular and Stroke Nursing, Council on Clinical Cardiology, and Council on Quality of Care and Outcomes Research. Guidelines for Adult Stroke Rehabilitation and Recovery: A Guideline for Healthcare Professionals From the American Heart Association/American Stroke Association. Stroke. 2016 Jun;47(6):e98-e169. doi: 10.1161/STR.0000000000000098. Epub 2016 May 4. Erratum in: Stroke. 2017 Feb;48(2):e78.
-Dąbrowski J, Czajka A, Zielińska-Turek J, Jaroszyński J, Furtak-Niczyporuk M, Mela A, Poniatowski ŁA, Drop B, Dorobek M, Barcikowska-Kotowicz M, Ziemba A. Brain Functional Reserve in the Context of Neuroplasticity after Stroke. Neural Plast. 2019 Feb 27;2019:9708905. doi: 10.1155/2019/9708905.
- Steinle B, Corbaley J. Rehabilitation of stroke: a new horizon. Mo Med. 2011 Jul-Aug;108(4):284-8.
4. In some places the use of English could be improved on.
Completing this gaps will have an impact on the understanding the aim of the study and, from my point of view, is absolutely necessary.
Author Response

(The authors gave the same response as above.)

Round 2
Reviewer 1 Report
The authors adequately addressed the issues.
Author Response
We appreciate your support.
Reviewer 2 Report
Dear Authors,
The changes and improvements made to the manuscript have greatly improved its scientific and formal quality.
However, there are two important aspects that I believe should be addressed:
1. The content checklist for systematic reviews in Rehabilitation should be the PERSIST not the PRISMA.
2. I think the Introduction does not emphasize enough the severity of stroke in terms of physical, psychological, emotional, family and socio-personal consequences and their treatment needs (e.g., doi: https://pubmed.ncbi.nlm.nih.gov/32806675).
Kind regards
Author Response
Comment 1: The content checklist for systematic reviews in Rehabilitation should be the PERSIST not the PRISMA.
Response) As reviewer’s suggestion, in the sport and exercise medicine, musculoskeletal rehabilitation and sports science fields for systematic reviews, the use of PERSiST(Prisma in Exercise, Rehabilitation, Sport medicine and SporTs science) is recommended in recent. We added this reference in main text, and the point where this part was not reflected was described in the limitations. JCM recommends following the PRISMA guidelines, and there is no mention of PERSIST in instruction for author.
Added reference:
- Ardern C.L,; Büttner F.; Andrade R.; Weir A.; Ashe M.C.; Holden S.; Impellizzeri F.M.; Delahunt E.; Dijkstra H.P.; Mathieson S.; Rathleff M.S. Implementing the 27 PRISMA 2020 statement items for systematic reviews in the sport and exercise medicine, musculoskeletal rehabilitation and sports science fields: the persist (implementing Prisma in exercise, rehabilitation, sport medicine and sports science) guidance. British journal of sports medicine 2022, 56(4),175-95. DOI:10.1136/bjsports-2021-103987
Comment 2: I think the Introduction does not emphasize enough the severity of stroke in terms of physical, psychological, emotional, family and socio-personal consequences and their treatment needs (e.g., doi: https://pubmed.ncbi.nlm.nih.gov/32806675).
Response) We briefly added the burden of stroke such as physical, psycho-social, emotional aspects in intro.
“Stroke has a negative impact not only on the individual's physical health, but also on the psychological, social, and emotional health, depending on the severity, and increases the burden on caregivers, including family members [1-4]”
